# Visualizing moiré ferroelectricity via plasmons and nano-photocurrent in graphene/twisted-WSe₂ structures

Shuai Zhang [1,9] ✉, Yang Liu [2,9], Zhiyuan Sun [3,8], Xinzhong Chen [1,4], Baichang Li [2], S. L. Moore [1], Song Liu[2], Zhiying Wang[2], S. E. Rossi[1], Ran Jing [1], Jordan Fonseca [5], Birui Yang[1], Yinming Shao [1], Chun-Ying Huang[6], Taketo Handa [6], Lin Xiong [1], Matthew Fu[1], Tsai-Chun Pan[1], Dorri Halbertal [1], Xinyi Xu[2], Wenjun Zheng [4], P. J. Schuck[2], A. N. Pasupathy [1], C. R. Dean [1], Xiaoyang Zhu[6], David H. Cobden [5], Xiaodong Xu[5], Mengkun Liu[4], M. M. Fogler[7], James C. Hone [2] & D. N. Basov [1] ✉

Ferroelectricity, a spontaneous and reversible electric polarization, is found in certain classes of van der Waals (vdW) materials. The discovery of ferroelectricity in twisted vdW layers provides new opportunities to engineer spatially dependent electric and optical properties associated with the configuration of moiré superlattice domains and the network of domain walls. Here, we employ near-field infrared nano-imaging and nano-photocurrent measurements to study ferroelectricity in minimally twisted WSe₂. The ferroelectric domains are visualized through the imaging of the plasmonic response in a graphene monolayer adjacent to the moiré WSe₂ bilayers. Specifically, we find that the ferroelectric polarization in moiré domains is imprinted on the plasmonic response of the graphene. Complementary nano-photocurrent measurements demonstrate that the optoelectronic properties of graphene are also modulated by the proximal ferroelectric domains. Our approach represents an alternative strategy for studying moiré ferroelectricity at native length scales and opens promising prospects for (opto)electronic devices.

Moiré superlattices of two-dimensional van der Waals materials have emerged as a capable platform to explore exotic electric and optical properties that can be controlled by selecting building blocks for the assembled atomic layers and manipulating the twist angles between them[1–3]. Moiré superlattices not only inherit characteristics of the constituent layers but also exhibit new emergent phenomena. Among these emergent effects, interfacial ferroelectricity was recently discovered in marginally twisted hexagonal boron nitride (h-BN)[4–7] and transition metal dichalcogenides (TMDs)[8–11]. In minimally twisted bilayers, lattice reconstruction results in triangular domains with periodically alternating AB and BA stacking registries[12–14]. The charge transfer between the two layers creates moiré patterns with alternating out-of-plane polarization[15,16]. Moiré ferroelectricity in twisted TMDs and h-BN has been confirmed by local Kelvin probe force

[1]Department of Physics, Columbia University, New York, NY 10027, USA. [2]Department of Mechanical Engineering, Columbia University, New York, NY 10027, USA. [3]Department of Physics, Harvard University, Cambridge, MA 02138, USA. [4]Department of Physics and Astronomy, Stony Brook University, Stony Brook, NY 11794, USA. [5]Department of Physics, University of Washington, Seattle, WA 98195, USA. [6]Department of Chemistry, Columbia University, New York, NY 10027, USA. [7]Department of Physics, University of California, San Diego, La Jolla, CA 92093, USA. [8]Present address: State Key Laboratory of Low-Dimensional Quantum Physics and Department of Physics, Tsinghua University, Beijing 100084, P.R. China. [9]These authors contributed equally: Shuai Zhang, Yang Liu. ✉e-mail: sz2822@columbia.edu; db3056@columbia.edu

microscopy (KPFM) imaging[5,7,10,17] and by area-averaged transport measurements[6,8].

Creating ferroelectricity by stacking vdW layers provides new opportunities to engineer materials and control their optoelectronic properties, as the ferroelectric polarization is expected to modulate the doping of an adjacent material[18] and tune photo-excited carrier dynamics[19], among other effects. In transport experiments, modulation of the doping of graphene is detected and employed as a sensor for the ferroelectric polarizations[6,8,20]. However, in transport experiments, the ferroelectrically induced doping density was inferred from area-averaged analysis by assuming that the entire sample reveals a uniform polarization under an applied electric field in a gatable device[6,8]. In practice, such devices may possess domains with the opposite polarization even at high electric fields, as was recently revealed by backscattering electron imaging[9] and KPFM[5]. This observation indicates that in stacked vdW devices, unlike conventional ferroelectric materials, aligning all the polarization in the same direction may require substantial energy for bending and eventually merging all the adjacent domain walls[9,21,22]. Therefore, in order to read out the doping from the ferroelectric potential, it is imperative to measure the carrier density in each moiré domain. Furthermore, novel optical and optoelectronic properties in devices integrated with ferroelectric have been investigated using far-field optical spectroscopy[23–26]. These measurements are diffraction-limited and thus cannot probe the rich optical or optoelectronic properties at the underlying moiré domain scale. Therefore, all existing results call for measurements capable of spatially resolving ferroelectric domains. Piezoresponse force microscopy (PFM) and KPFM can directly measure the domain structures of ferroelectric via electromechanical surface deformation and electrostatic force, respectively. Whereas the ferroelectric domains can be clearly visualized by PFM[27], PFM results directly reflect the piezoelectric response, making it challenging to quantitatively evaluate ferroelectric properties. Usually, KPFM can provide a quantitative characterization of a ferroelectric by measuring the work function[5,7]. But KPFM has difficulty in quantitatively characterizing ferroelectric devices that include multiple materials because disentangling the electrostatic force from individual layers is a formidable task.

Here, we utilize near-field infrared nano-imaging and nano-photocurrent to investigate the optical and optoelectronic properties of back-gated graphene encapsulated with a minimally twisted bilayer of WSe$_2$ (t-WSe$_2$) revealing ferroelectricity. We first demonstrate that the plasmonic response of graphene is altered by the proximate twisted ferroelectric domains. Notably, by investigating the local plasmonic contrast in graphene residing underneath the ferroelectric domains, we obtain a reading of the local ferroelectric polarization. Moreover, we show that the proximity to the ferroelectric bilayer breaks the inversion symmetry and modulates the Seebeck coefficient of graphene, thereby allowing the generation of photocurrent. The nano-photocurrent mapping displays moiré patterns and further confirms the notion of the local modulation of the carrier density in graphene prompted by ferroelectric domains. These results uncover alternative approaches to controlling the optoelectronic response of graphene integrated with ferroelectric materials.

## Results and discussion
### Device structure and ferroelectric doping
We investigated a series of graphene/t-WSe$_2$ devices with the same general architecture. These devices are based on back-gated graphene structures, with graphene encapsulated by a minimally twisted bilayer of WSe$_2$ on the top and h-BN at the bottom (Fig. 1a). The t-WSe$_2$ was assembled from the same microcrystal of monolayer WSe$_2$, which was first cut into two halves by laser ablation and then assembled using a dry stacking process without any intentional rotation[28]. The entire stack benefits from a global back gate with the h-BN and 285-nm SiO$_2$ together constituting the gate dielectric. The graphene layer has

several electrical contacts, enabling its electrostatic gating as well as nano-photocurrent measurements[29–32].

The experimental concept is outlined in Fig. 1b. The ferroelectric polarization in a given domain of t-WSe$_2$ gives rise to an electrical potential near its surface[5,7,33]. This potential alters the local carrier density and hence the local Fermi energy of an adjacent graphene layer[8] (Fig. 1b). The ferroelectrically induced carrier density in graphene had been demonstrated by putting graphene on an oxide ferroelectric material[34]. In addition, recent electrical transport measurements also indicated that graphene is doped by the adjacent two-dimensional ferroelectrics[6,8]. The resultant carrier density can be quantified by interrogating the plasmonic response of graphene. In our infrared nano-spectroscopy experiments, the plasmonic response is manifest in two complementary observables: (i) the magnitude of the near-field scattering signal[35] and (ii) the periodic oscillating patterns (fringes) arising from the propagating plasmon polaritons[36,37]. In principle, both observables are governed by the carrier density in graphene. In practice, for samples with moiré super-structures, all domain boundaries are potentially capable of launching and reflecting propagating plasmons[38], leading to complex patterns that make it difficult to extract the local carrier density. Therefore, we will primarily focus on an analysis of the evolution of the near-field amplitude $s(n,\omega)$ in the following sections, where $n$ is the carrier density and $\omega$ is the photon energy. To guarantee the accuracy of the local near-field amplitude and the corresponding carrier density, we should ensure that propagating/reflected plasmon polaritons do not contribute to the scattering signals. An in-depth discussion of quantifying carrier density based on nano-infrared studies of graphene can be found in Supplementary Note 1.

### Infrared nano-imaging of plasmonic response of t-WSe2/ graphene
A representative image of the scattering amplitude acquired with a photon energy of 880 cm$^{-1}$ is shown in Fig. 1c. From this image, we can clearly see moiré patterns. The moiré structures display nearly triangular domains, which result from the lattice reconstruction[13,39]. The period of the moiré patterns is inhomogeneous as the twist angle between the WSe$_2$ layers varies over the field of view due to strain and wrinkles. Indeed, across the wrinkles, the observed moiré periods prominently change. Intriguingly, the nearest-neighbor domains reveal clear contrast, while the next-nearest-neighbor domains show almost the same near-field intensity. We performed measurements for several devices, all of which consistently produced these moiré features (Supplementary Note 2 and Supplementary Fig. 1 and 2).

Now we inquire into the origin of the observed contrast between adjacent domains. For this purpose, we varied the carrier density in the graphene layer in our back-gated devices. Representative images acquired at various gate voltages are shown in Fig. 1d (a full set of images is plotted in Supplementary Figs. 3 and 4). From these images and the line profiles in Fig. 1e, we can see that the contrast between moiré domains systematically evolves as a function of the carrier density. Moreover, at high doping with $V_g - V_{CNP} \geq 65\,V$, the propagating plasmon polaritons manifest as fringes of nano-infrared contrast and can be observed near the boundaries of large domains (Supplementary Fig. 3g–i). At all of these back-gate doping regimes, the contrast between the domains can be consistently observed. It has been well established that plasmon of graphene can be tuned by carrier density, thus enabling the observation of propagating plasmon polaritons[35–37]. Therefore, both the observed carrier density-dependent contrast and the plasmonic fringes attest to the conclusion that the ferroelectric polarization modulates the graphene plasmons and gives rise to moiré patterns in the nano-infrared images.

We stress that the contrast observed in Fig. 1 originates from the plasmonic response of graphene rather than that of WSe$_2$. The WSe$_2$ layers are not doped by the back gate in our experiment as they are not

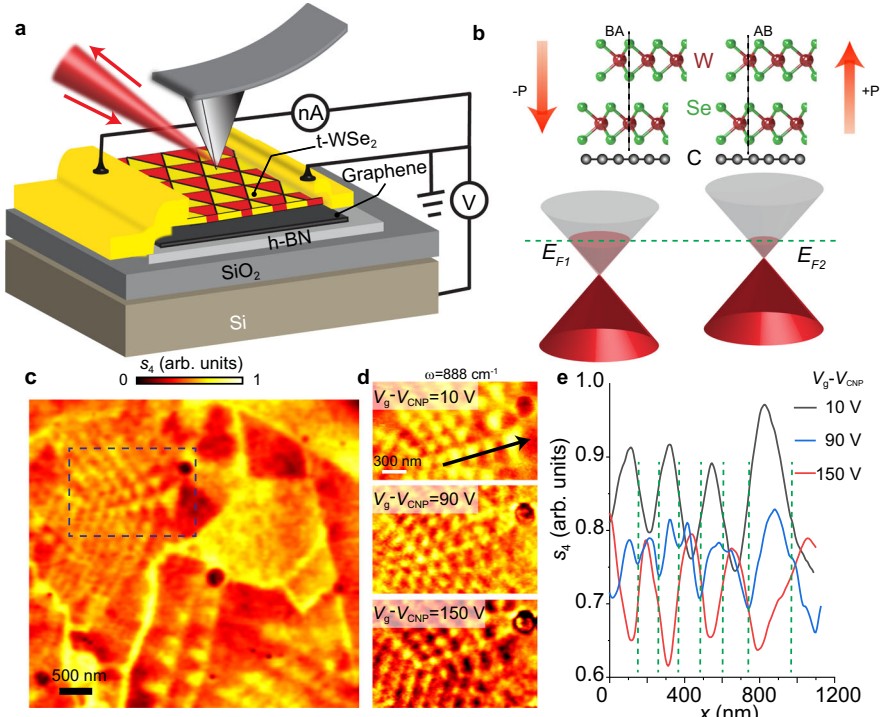

**Fig. 1 | Moiré ferroelectricity investigated via near-field infrared nano-imaging.** **a** Schematics of the back-gated graphene device with a minimally twisted WSe$_2$ bilayer (t-WSe$_2$) on top. This structure is suitable for simultaneous near-field scattering and nano-photocurrent experiments. **b** Top: side view illustration of device structures. AB and BA stacking registries are formed in the R-stacked domains in bilayer WSe$_2$. The two distinct out-of-plane atomic alignments give rise to opposite out-of-plane polarizations. Bottom: the Fermi energies $E_{F1}$ and $E_{F2}$ of graphene are modulated by the alternating ferroelectric polarization of moiré domains in t-WSe$_2$. **c** Near-field scattering amplitude mapping of Device A carried out with photon

energy $\omega = 880$ cm$^{-1}$ and $V_g - V_{CNP} = 10$ V, where $V_{CNP}$ is the moiré-averaged charge neutrality point defined in the text, and $V_g$ is the back gate voltage. The image clearly exhibits the moiré superlattices formed in the minimally twisted WSe$_2$. The near-triangular domains are formed due to the lattice relaxation. **d** Nano-infrared images acquired for different gate voltages with excitation energy $\omega = 888$ cm$^{-1}$. **e** Line profile of the scattering amplitude, $s_4$, for selected back gate voltages measured along the line indicated in Panel d. Green dashed lines denote the positions of the domain walls. All data were acquired at $T = 300$ K for Device A.

subjected to the electric field confined between graphene and the back gate[40]. The Drude response of WSe$_2$, which potentially could originate from extrinsic doping, does not extend to mid-IR frequencies because of the relatively high effective mass of either electrons or holes in WSe$_2$[41]. (Supplementary Note 4 and Supplementary Figs. 7,8,9).

**Mapping the local carrier density in ferroelectric moiré domains**
In the previous section, we established that the ferroelectric potential can dope proximal graphene and that the doping density can be monitored by examining the local plasmonic response of graphene. Now we proceed to extract the magnitude of the local domain-dependent doping, which will provide nanoscale maps of the ferro-electric polarization. In our structures, graphene is doped by both the ferroelectric potential and the electrostatic gating of the back gate. Naturally, the global back gate induces a spatially uniform carrier density across the entire graphene microcrystal. On the other hand, the alternating ferroelectric polarization in the AB and BA domains of t-WSe$_2$ results in the modulation of the local carrier density, leading to marked contrast between the neighboring domains (Fig. 1c, d and Fig. 2c, d). We are in a position to disentangle the ferroelectric and back gate contributions by examining the local variations of the plasmonic response. Specifically, we will examine the back gate dependence of the nano-IR contrast.

The near-field scattering amplitude, which reveals plasmon evolution, was measured as a function of both spatial position and back gate voltage. Representative data acquired for Device B are displayed in Fig. 2. In the course of these measurements, the tip was repeatedly scanned along the same line across three ferroelectric domains (blue arrow line in the inset of Fig. 2c) while the back-gate voltage was

gradually swept from −70 V to 70 V; the scattering amplitude was recorded, forming a 2D coordinate-voltage plot in Fig. 2a. To display the near-field amplitude evolution more clearly, two line cuts extracted from the centers of the AB and BA domains are plotted in Fig. 2b. The non-monotonic evolution of the scattering amplitude is evident in the data presented in Fig. 2a, b.

The data displayed in Fig. 2 show that the graphene in the AB and BA domains reaches charge neutrality point (CNP) at distinct back gate voltages, due to the presence of carriers from opposite ferroelectric polarizations. For each domain, at the CNP, the near-field scattering amplitude reaches a local maximum. With doping, the near-field amplitude first decreases and then increases, forming a V-shaped spectrum. Combining the electron and hole doping, a W-shaped scattering amplitude profile is formed (Fig. 2b). This trend is further confirmed by numerical simulations and analytical calculations (Supplementary Note 3, Supplementary Fig. 6). The near-field amplitude evolution can be understood by employing the Fresnel reflection coefficient of p-polarized light, $r_p(\omega,q)$, which is a function of the plasmon polariton energy, $\omega$, and momentum, $q$, and governs the near-field amplitude. At low doping, $Amp(r_p(\omega,q))$ decreases with doping, and thus the near-field amplitude decreases. When the carrier density, $n$, is further increased, the plasmon polariton momentum gradually decreases and finally matches the tip momentum. Consequently, $Amp(r_p(\omega,q))$ surges tremendously, resulting in an upturn of the near-field amplitude.

Around the CNP, the two scattering amplitude profiles acquired on the AB and BA domains show a shift of $\Delta V_g = 12$ V (Fig. 2b). This shift indicates that a corresponding doping from the back gate should be supplied to one domain in order to compensate for the doping

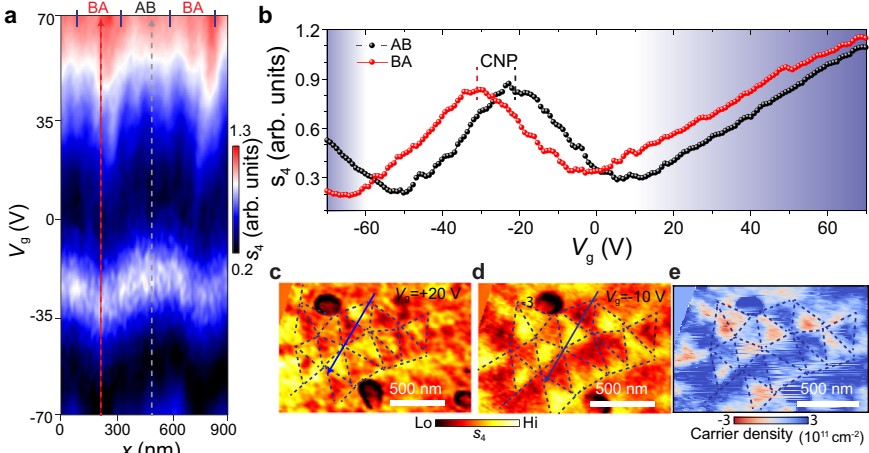

**Fig. 2 | Visualizing and quantifying the plasmonic response of individual ferroelectric domains in t-WSe₂. a** The fourth harmonic of the near-field scattering amplitude, $s_4$, plotted as a function of gate-voltage $V_g$ and measured for a line trace that crosses three ferroelectric domains (see blue arrow line in Panel **c**). The positions of the domain wall are indicated on the top by four solid lines.
**b** Representative results for the voltage-dependence of the scattering amplitude, $s_4(V_g)$, probed within AB and BA domains. These two traces were extracted from data in Panel **a** (dashed lines). The charge neutrality points (CNP) are distinct for the AB and BA domains. In the shaded regions, graphene is heavily doped and supports propagating plasmon polaritons launched by the tip. **c, d** Images of the near-field scattering amplitude, $s_4$, acquired at $V_g = +20\,V$ and $V_g = -10\,V$, respectively. The dashed lines mark the domain boundaries. These images reveal the contrast reversal between AB/BA domains. **e** Carrier density mapping extracted from the near-field amplitude $s_4$ as described in the text. All the data were acquired with photon energy $\omega = 862\,cm^{-1}$ and, for Device B, with a moiré-averaged CNP of about $-26\,V$.

difference from opposite ferroelectric polarizations. Based on the device geometry, we could translate the voltage into the carrier density difference between adjacent domains, which is $7.5 \times 10^{11}\,cm^{-2}$. With the moiré-averaged Fermi energy at charge neutrality, this doping density shifts the Fermi energy (Fig. 1b), measured with respect to the Dirac point in the two ferroelectric domains, by $\pm 71\,meV$. This value is about four times higher than the theoretically predicted 16 meV, based on a linear screening model with a ferroelectric potential of 56 mV (Supplementary Note 5)[39,42]. The unexpected high doping implies that the ferroelectric potential is not the sole mechanism responsible for generating the carrier density in graphene. The interfacial charge trapping and defect states in WSe₂ or h-BN are potential mechanisms acting concomitantly with the ferroelectric polarization (Supplementary Notes 5 and 10). In our device, graphene is encapsulated by WSe₂ and h-BN. Defect states in WSe₂ and h-BN could result in charge transfer between WSe₂ (or h-BN) and graphene, a process governed by the Fermi energy[43]. (Defects in h-BN and WSe₂ can be confirmed by optical measurements in Supplementary Note 10 and Supplementary Fig. 17). Therefore, the ferroelectrically tuned Fermi energy of graphene may give rise to unequal charge transfer in the two neighboring domains. As a result, charge transfer can enhance the graphene doping beyond the ferroelectric potential (see Supplementary Note 5.5). The residuals at the interface can also enhance the doping contrast by modifying dielectric screening, but they play only a minor role (Supplementary Note 9). We remark that the reported ferroelectric potentials/doping in existing transport results is consistent with first-principle calculations;[6,8] implicit in this latter analysis is a hypothetical assumption that the entire device undergoes polarization switching. Our experiments suggest that a quantitative analysis of ferroelectricity in moiré structures needs to combine transport and nano-imaging experiments.

It is noteworthy that, except for the regime with propagating plasmons (see the shaded areas in Fig. 2a and further discussion in Supplementary Notes 7,8), the voltage shift between the two scattering amplitude profiles stays constant (Fig. 2b). This invariable ferroelectrically induced doping contrast indicates a constant ferroelectric polarization amplitude when graphene is doped. Therefore, the ferroelectric polarization originating from charge transfer between WSe₂ layers is robust and is immune from electron screening of carriers in nearby graphene. From Fig. 2b, the relation between the carrier density and the near-field amplitude is established. Then, the scattering amplitude images allow us to map out the carrier density induced by the ferroelectric polarization, as shown in Fig. 2e. All the data in Fig. 2 demonstrate that plasmonic response reveals the ferroelectric polarization and enables the quantification of the ferroelectric-induced carrier density.

While the back gate voltage is being tuned, the near-field amplitude contrast between AB and BA reverses three times, as shown by the line profiles in Fig. 2b. The contract reversal can also be visualized by comparing images acquired at various back gate voltages (Fig. 1d and Fig. 2c, d). These contrast reversals originate from the non-monotonic evolution of the plasmonic signal with doping rather than from polarization switching. This non-monotonicity can also induce period doubling (middle panel of Fig. 1d and Fig. 1e). In our structures, the ferroelectric WSe₂ residing above the graphene is intentionally not subjected to the electric field, excluding the possibility of polarization switching.

## Ferroelectrically engineered photo-thermoelectric effect

Now we use nano-photocurrent imaging to study how the moiré ferroelectricity of t-WSe₂ controls optoelectronic properties of graphene. The photocurrent imaging modality is a readily available modality of scattering-type scanning near-field optical microscope (s-SNOM) experiments[29,30,44,45]. Nano-photocurrent imaging has been extensively used to investigate domains and domain walls in twisted bilayer graphene[31,32,46]. The formation of photocurrent requires broken inversion symmetry, which can be accomplished by introducing p-n or n-n⁺ junctions in the graphene layer. In our devices, the staggered ferroelectric potential breaks the inversion symmetry, thereby allowing photocurrent formation.

The nano-photocurrent mapping in Fig. 3a clearly shows that the graphene/t-WSe₂ displays moiré patterns marked by prominent sign flipping: red and blue colors denote positive and negative currents, respectively. In addition to the moiré regions, the photocurrent flips sign across wrinkles and terrace edges, as shown in Fig. 3b. Near all boundaries and edges, the photocurrent amplitude exhibits a gradient. This photocurrent gradient blurs moiré patterns, in contrast to the much sharper appearance of domains and boundaries revealed by the scattering amplitude images (Fig. 3c, d). Also, by comparing the

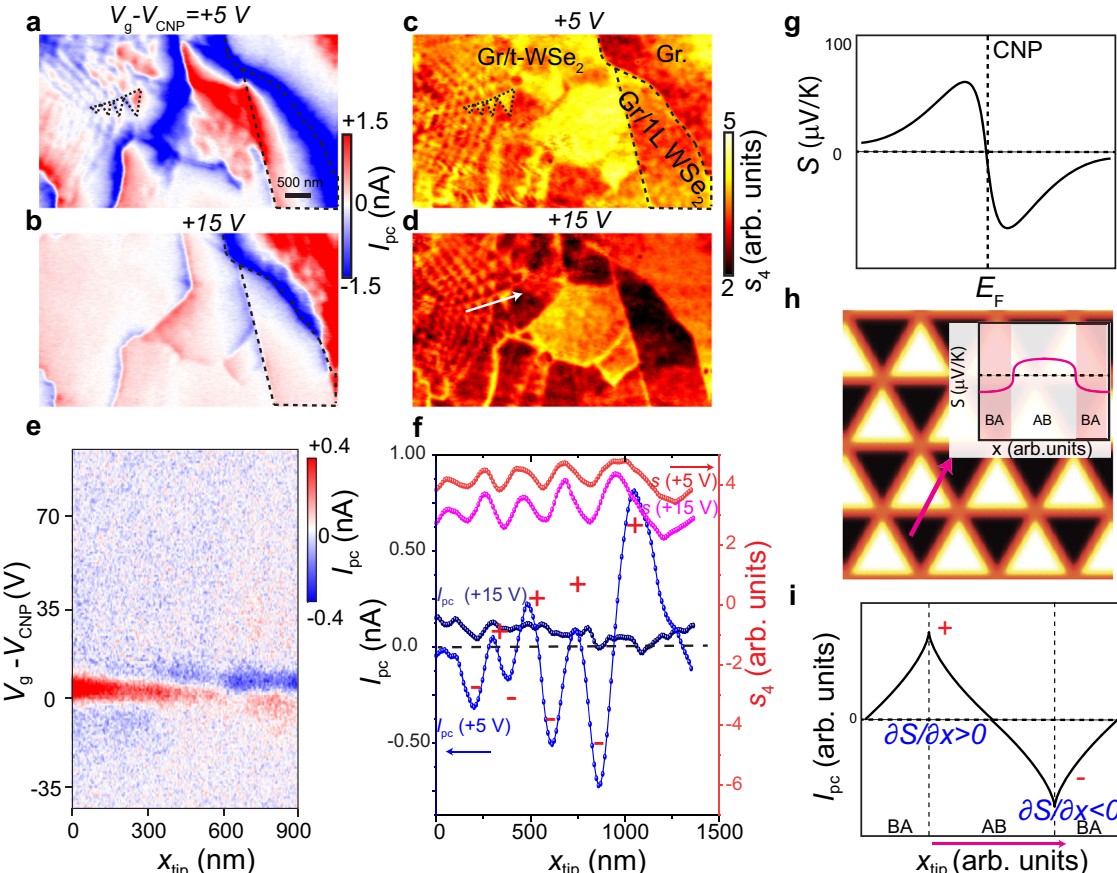

**Fig. 3 | Nanoscale photo-thermoelectric response of graphene modulated by moiré ferroelectricity in proximal t-WSe₂.** **a**, **b** Images of the gate-controlled photocurrent ($I_{pc}$) acquired at excitation energy $\omega = 880\ cm^{-1}$. The back gate voltage is denoted above each image. **c**, **d** Images of the near-field scattering amplitude $s_4$ simultaneously acquired with **a** and **b**, respectively. The scale bar in **a** also applies to Panels **b-d**. The dashed lines in **a** and **c** mark domain boundaries. **e** The photocurrent as a function of gate voltage, $V_g - V_{CNP}$, measured along the blue line in Fig. 2c. **f** Line profiles of the photocurrent $I_{pc}$ and of the scattering amplitude $s_4$ for several back gate voltages. Data were collected at the location of the white arrow in **d**. The photocurrent acquired at $V_g - V_{CNP} = 5\ V$ displays sign flipping. **g** The Seebeck coefficient of graphene as a function of Fermi energy. **h** The Seebeck coefficient forms a checkerboard pattern. Thus, a photocurrent can be formed. Inset: The Seebeck coefficient profile, along the pink line, in graphene engineered by adjacent ferroelectric AB and BA domains. A nonzero gradient of the Seebeck coefficient is formed at the domain wall. **i** The photocurrent forms near the domain boundary due to the Seebeck coefficient gradient (schematic). The neighboring boundaries display sign flipping of the photocurrent, which originates from the opposite signs of the Seebeck coefficient gradient. All the photocurrent data were acquired by demodulation at the second harmonic of tip tapping frequency. Data in (**e**) were acquired for Device B. All other data were acquired for Device A. For Device A, we measured a series of photocurrent/nano-IR images, with a back gate voltage step of 5 V. We collected a detailed series of images for this device, yet the back gate dependence data are only fragmentary. Thus, we acquired this additional information for Device B. The data presented for Device B take the form of a line scan, with a fine back gate step size of 0.8 V. In order to present more comprehensive photocurrent evolution data, data acquired on Device B are used in (**e**).

photocurrent image in Fig. 3a with that in Fig. 3b, we find that the photocurrent nearly vanishes when the graphene Fermi energy is tuned far away from the CNP. The doping dependence and the spatial gradients suggest that the photocurrent arises from the photo-thermoelectric effect (PTE)[47,48].

Now we elucidate how the PTE generates the photocurrent and interpret the above photocurrent features. In our experiment, when the tip is scanned over the devices, the tip-focused electric field increases the electron temperature of graphene. As a result, local thermoelectric current **j** is formed, $\mathbf{j} = \sigma S \nabla \delta T$, where $\sigma$ is the DC conductivity, $S$ is the Seebeck coefficient, and $\delta T$ is the increased electron temperature. The current collected by the contacts can be calculated by invoking the Shockley-Ramo theorem[49], $I_{PC} = \int \mathbf{j} \cdot \nabla \psi d^2\mathbf{r}$, where $\psi$ is an auxiliary field obtained by assigning potentials of 1 and 0 to contacts. Taken together, the collected photocurrent is expressed as:

$$I_{pc} = \int_{\Omega} \sigma S \nabla \delta T \cdot \nabla \psi d^2\mathbf{r} = \int_{\partial\Omega} \sigma S \delta T \nabla \psi \cdot d\hat{\mathbf{n}} - \int_{\Omega} \sigma \delta T \nabla \psi \cdot \nabla S d^2\mathbf{r} \quad (1)$$

The first term on the right-hand side of Eq. (1) denotes the photocurrent formed at the contact edges. When the tip is far away from the contacts, farther than the cooling length, the local field underneath the tip cannot heat the electrons at the contact ($\delta T = 0$), resulting in zero photocurrent. So the first term can be safely discarded when the tip is far away from the contacts. Thus, the photocurrent is dominated by the second term, showing that the photocurrent can be detected if a device has a nonzero Seebeck coefficient gradient, $\nabla S$, along the auxiliary field.

The Seebeck coefficient of graphene depends on the Fermi energy (Fig. 3g). As the Fermi energy is modulated in ferroelectric domains, the Seebeck coefficient forms a triangular checkerboard pattern (Fig. 3h). Therefore, a nonzero Seebeck coefficient gradient is formed at the domain boundaries and results in photocurrent. Near the CNP, the Seebeck coefficient gradient between AB and BA domains reaches a maximum, and thus the maximum photocurrent forms (Fig. 3e). When the doping is increased via a back gate, the Seebeck coefficient contrast between AB and BA domains diminishes. As a result, photo-current fades out at high doping (Fig. 3e).

With the photocurrent mechanism in mind, we can gain a deeper understanding of the spatial features of the photocurrent. When graphene is locally heated, the heat spreads on a characteristic length referred to as the cooling length, which is hundreds of nanometers (Supplementary Note 11). Thus, as long as the distance between the tip and the domain boundaries is shorter than the electron cooling length, the electronic temperature of the domain boundaries will rise and the photocurrent will be detected. When the tip gradually moves away from the boundaries, $\delta T$ at the boundaries reduces. The resultant photocurrent decays as a function of distance from the boundaries (Fig. 3a, f). In addition, the neighboring domain walls possess Seebeck coefficient gradients with opposite signs, which results in sign flipping of the photocurrent (Fig. 3f, i). In contrast to infrared scattering spectra, which originate from regions right underneath the tip, the thermoelectric photocurrent is contributed to by regions determined by the cooling length. As a result, whereas the observed photocurrent patterns resemble those in scattering images (Fig. 3c, d), the photocurrent exhibits a more complicated texture with blurred moiré patterns. All of these photocurrent features are well reproduced by simulations (Supplementary Note 12).

Our analysis of the nano-photocurrent not only confirms that the graphene Fermi energy is modulated by the proximal ferroelectric but also provides a method for studying the ferroelectric-engineered optoelectronic properties. We note that the inversion symmetry breaking of graphene, a prerequisite for photocurrent generation, is governed by the proximity effect. Thus, the inversion symmetry of graphene can be controlled by switchable ferroelectricity. Notably, the width of the p-n junction formed by ferroelectric doping is ~10 nm, which is much narrower than the depletion region in conventional p-n junctions and enables emerging applications, such as efficient energy harvest[19] and miniaturized sensors for electric fields.

To summarize, the ferroelectric domains in minimally twisted WSe₂ were visualized by examining the plasmonic response in the proximal graphene monolayer. The analysis of the plasmonic data allows us to infer ferroelectric polarization in t-WSe₂. Complementary nano-photocurrent measurements demonstrate that the moiré ferroelectricity can tune the optoelectronic properties of graphene. Plasmonic sensing of ferroelectricity established through our experiments is readily applicable to the analysis of other synthetic vdW ferroelectrics[11,18,50,51]. Integrating the graphene layer with a ferroelectric allows us to explore how the ferroelectric tunes the properties of the proximal materials, which is crucial for the application of ferroelectric materials. We note that this proximal material is not limited to graphene. Recent advances in the nano-spectroscopy/imaging of excitonic effects in vdW materials[52,53] set the stage for the analysis of the impact of ferroelectricity on the excitonic resonance energy and exciton diffusion[25], and for photovoltaic applications[23]. Compared to PFM and KPFM, which are surface-sensitive methods, s-SNOM is a complementary method with the capability for tomographic imaging[54]. Therefore, a s-SNOM enables one to probe hidden interfaces in devices with top electrodes, for example. In the immediate future, our demonstrated method can also be used to visualize the evolution of ferroelectric domains under uniform electric fields in practical devices. This unique capability is essential for understanding the fundamental dynamics of polarization.

## Methods
### Device fabrication
Monolayer WSe₂ and few-layer h-BN were cleaved on 285-nm SiO₂ substrates using the typical Scotch tape method. The thickness was identified via optical microscopy and re-confirmed by Raman microscopy (Renishaw Raman system) and the use of a Bruker atomic force microscope (AFM). Monolayer WSe₂ was first cut into two halves using a laser cauterization method and then underwent a dry stacking process in which an h-BN/graphene stack was used to pick up the two

pieces of identical WSe₂ without any intentional rotation. The final stack was flipped upside down to expose the WSe₂/graphene upward using a flipped chip method[55]. Finally, we used low-temperature, high-vacuum annealing to remove the buried polypropylene carbonate (PPC). The final device geometry was defined by electron-beam lithography and reactive ion etching (RIE, Oxford Plasmalab 100 ICP-RIE instrument), followed by electron-beam evaporation (EBE) to deposit Cr/Au = 5/150 nm as the surface contacts. Piezoresponse force microscopy (PFM) imaging was achieved with the Bruker atomic force microscope.

### Nano-infrared scattering experiments
The nano-infrared scattering experiments were performed using a home-built scattering-type scanning near-field optical microscope (s-SNOM) housed in an ultra-high vacuum chamber with a base pressure of ~$7 \times 10^{-11}$ torr. The s-SNOM is based on a tapping-mode atomic force microscope (AFM). The tapping frequency and amplitude of the AFM are about 75 kHz (or 285 kHz) and 70 nm, respectively.

The s-SNOM works by scattering tightly focused light from a sharp AFM tip. The spatial resolution of the s-SNOM is predominantly set by the tip radius of curvature $a$ (~20 nm in our experiment) and therefore allows us to resolve optical and optoelectronic properties at a length scale below that of the ferroelectric domains. First, the laser was focused on a metalized AFM tip using a parabolic mirror. Then the back-scattered light was registered. With this approach, we obtained the genuine near-field signal at a resolution of ~20 nm. In addition to the scattering signal, photocurrent was simultaneously recorded and demodulated at high harmonics of the tip tapping frequency, yielding nano-photocurrent.

The tapping amplitude is used as the z feedback. The tip excitation phase is controlled by a phase lock loop. The AFM also has the functions of a side-band Kelvin probe force microscope, and the feedback bias voltage is applied onto the AFM tip to compensate for the potential difference between the tip and the sample. This applied bias can ensure that the tip vibration is not influenced by the ferroelectric potential. Thus, the observed near-field contrast is the genuine near-field scattering signal, which is generated purely from the sample conductivity. During the near-field mapping, the z feedback, phase lock loop, and Kelvin probe force microscope are turned on simultaneously. We note that for a tip with a resonance frequency of 285 kHz, the large force constant of about 42 N/m makes it insensitive to the potential difference between the sample and the tip, so no discernible potential response is detected by the tip. Therefore, the KPFM function is not required when using tips with a stiff cantilever.

The laser source is a tunable quantum cascade laser (QCL) from Daylight Solutions. Photon energy of 860 - 920 cm⁻¹ was used to avoid phonon resonances from substrate h-BN and SiO₂. The laser beam was focused onto the metallized AFM tip using a parabolic mirror with a 12-mm focal length. The back-scattered light was registered by a mercury cadmium telluride (MCT) detector and demodulated following a pseudoheterodyne scheme. The signal was demodulated at the *nth* harmonic of the tapping frequency, yielding background-free images. To eliminate the far-field background, we chose $n = 3$ and 4 in this work.

### Nano-photocurrent experiments
The nano-photocurrent measurements were performed in a home-built s-SNOM housed in an ultra-high vacuum chamber with a base pressure of ~$7 \times 10^{-11}$ torr. Unless otherwise stated, the laser source was a tunable QCL from Daylight Solutions. The laser power was set to be ~10 mW. The current was measured using a current amplifier with a gain setting of $10^7$ and corresponding bandwidth >1 MHz. In order to isolate the photocurrent generated by the near fields underneath the tip, the photocurrent was sent to a lock-in amplifier and demodulated at the *nth* harmonic of the tip tapping frequency. In this work, $n = 2$ was used.

## Data availability

The raw data in the current study are available from the corresponding authors upon request.

## Code availability

The code used for the analysis and simulations in the current study is available from the corresponding authors on request.

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

## Acknowledgements
Nano-imaging research at Columbia is supported by DOE-BES Grant No. DE-SC0018426. Research at Columbia on moiré superlattices is entirely supported as part of Programmable Quantum Materials, an Energy Frontier Research Center funded by the U.S. Department of Energy (DOE), Office of Science, Basic Energy Sciences (BES), under Award No. DESC0019443. D.N.B. is a Moore Investigator in Quantum Materials EPIQS GBMF9455. Device fabrication and crystal growth (Y.L., B.L., S.L., Z.W., J.C.H.) are supported by the NSF MRSEC program at Columbia through the Center for Precision-Assembled Quantum Materials (DMR-2011738).

## Author contributions
D.N.B. conceived the study. S.Z. conducted the nano-IR measurements. S.Z. built the nano-IR instruments with help from L.X., M.F and T.-C.P. Y.L., supervised by J.C.H., fabricated the devices used in the main manuscript and conducted the PFM measurements. B.L., Z.W. and S.L., supervised by J.H., fabricated multiple devices at the early stage of this work. X.C. and W.Z., supervised by M.L., performed the near-field scattering simulations. S.E.R., D.H., R.J. and S.Z. performed the photocurrent simulations. S.L., supervised by J.C.H., grew the crystals. C.-Y.H. and T.H., supervised by X.Z., performed the Raman measurements. Z.S. and M.F. provided theoretical modeling of ferroelectric doping. Xia.X., D.H.C., A.N.P., S.L.M., S.Z. and D.N.B. analyzed the ferroelectric doping data. J.F. and S.L.M. fabricated devices to confirm the conclusion of ferroelectric-induced doping. B.Y. supervised by C.R.D., fabricated the devices used to measure the graphene plasmon evolution. Xin.X., supervised by P.J.S., fabricated some devices for KPFM measurements. Y.S. performed KPFM measurements of some devices. S.Z. and D.N.B. wrote the manuscript with input from all co-authors.

## Competing interests
The authors declare no competing interests.
