## [Peer Review File · Nature Communications]

Visualizing moiré ferroelectricity via plasmons and nano-photocurrent in graphene/twisted-WSe₂ structuresREVIEWER COMMENTS

Reviewer #1 (Remarks to the Author):

In the study the authors demonstrate the nano-optical visualization of moiré domains in a twisted-WSe₂ bilayer. By placing a layer of graphene on top of the twisted-WSe₂ bilayer (TWB), they are able to observe distinct ferroelectricity in two different TWB domains using near-field optical microscopy. This is possible because the carrier densities, which determine the plasmonic responses, of graphene above the two distinct TWB domains differ due to the opposite electric polarizations in these ferroelectric domains. The authors present an intriguing idea and provide comprehensive measurements and calculations to support their conclusions, offering a new perspective and approach for studying nanoscale ferroelectricity in similar systems. I think the manuscript is interesting and timely. Here are a few concerns that I believe should be addressed by the authors prior to the publication of this manuscript.

1) This ferroelectric can be detected and imaged with other scanning probe microscopy such as piezoelectric force microscopy (PFM). What is the advantage of the method proposed in this work when compared with PFM? Introducing a graphene layer and back-gate to the bilayer and then image the sample with s-SNOM indeed can help obtain the local carrier density of different TWB domains, but in the meantime may also change the stacking order or twist angle of these domains (induced by strain effect when heating or annealing the sample) and also introduce other disturbing factors such as residues between different layers. The novelty of the current study should be stated more clearly.

2) The measured carrier density (or Fermi level) of graphene is a key parameter for drawing conclusions in this paper. Therefore, the accuracy of deriving this number is of utmost importance. In the s-SNOM imaging, the domain wall of the twisted WSe₂ can reflect the tip-launched plasmon, and the domain sizes are comparable to the wavelength of graphene plasmons. Consequently, the experimentally obtained scattering amplitude may arise from the overall contribution of locally excited and reflected plasmons, similar to those in a graphene plasmonic cavity. This scenario could potentially interfere with the modulating of the doping level of graphene. How did the authors address or eliminate this possibility?

3) The authors should state more clearly on how to obtain the carrier density in graphene at different gate voltage.

4) In line 204, they wrote, 'but the ferroelectric polarization tuned Fermi energy makes these processes behave differently on two adjacent domains,' without specifying the exact different behavior exhibited by the two adjacent ferroelectric domains. Additionally, they cite Ref 41, which reported the visualization of defects on boron nitride, to support this statement. Please check if this is an appropriate citation.

5) In line 190, Page 7, they wrote "This is about four times higher than the theoretical prediction based on a linear screening model with ...". Is that a typing mistake? I could not find the exact information about this quantity relation, the authors are suggested to explain the statement more specifically.

6) In equation (1) the authors neglect the first term on the right-hand side, without explaining the reason.

Reviewer #2 (Remarks to the Author):

In this paper, the authors studied ferroelectricity in minimally twisted WSe₂ by using near-field infrared nano-imaging and nano-photocurrent measurements. They claimed that the plasmonic response in a graphene layer adjacent to the moire WeS₂ bilayers could be used to unveil the local ferroelectric polarization. The proposed technique is able to resolve ferroelectric domains in nanoscale, which should be potentially useful for nanocale ferroelectricity research for optoelectronic devices. The paper is well-written and could be considered publication in Nat.

Commun. as long as following issues are considered.

1. First, the near-field amplitude instead of plasmon wavelength is used to determine the carrier density of the graphene in this work. However, the near-field amplitude is affected by lots of facts, such as the tip, laser intensity and so on. Could the authors comments on further how to guarantee the quantized accuracy?
2. In Fig.3, some data comes from Device B instead of Device A, what's the difference in physical properties between Device A and B? Besides, it is a bit confused to compare the result comes from different samples.
3. The infrared laser with a frequency of 880 cm^{-1} is used in this work, is there any special reason for the choice?
4. In Fig.1e, an obvious different variation trend exists for the case of 90V compared with that of 10V and 150V, what's the possible reason for the difference?
5. Due to the complex doping mechanism for the graphene, a quantitative measurement of the ferroelectricity of the twisted WSe₂ may needed, which can verify the correctness of the explanation for experimental results.

Reviewer #3 (Remarks to the Author):

The authors reported the observation of ferroelectric domains in twisted WSe₂ using near-field infrared nano-imaging and nano-photocurrent techniques. This is achieved because of the fact that the plasmonic response of monolayer graphene depends on its carrier density, which is turned by the polarization of the twisted WSe₂. The work is of interest to a large audience, and I recommend publication of the manuscript after the following minor concerns are clarified.

- 1, Regarding Fig. 2b, a qualitative description in the main text on why would the near-field amplitude first decreases and then increases would be helpful for the general audience.
- 2, Regarding the contrast between neighboring domains, the authors invoke interfacial charge trapping and defect states in WSe₂ or h-BN to explain the larger-than-expected contrast. This is a bit confusing. If one assumes these trappings or defects distribute uniformly across the samples, shouldn't they contribute similarly to the backgate, as in shifting the CNP of both domains in the same direction? Can the authors elaborate on how would they increase the contrast between domains?
- 3, Regarding the photocurrent, due to the ferroelectric polarization, the Fermi levels of the graphene beneath neighboring domains are different and a potential difference naturally develops across the domain wall, will this lateral potential variation produce photocurrent? Is it absolutely necessary to introduce Seebeck effect?
- 4, How large is the laser spot as compared to the moire pattern in the nano-photocurrent measurements?

We thank the reviewers for recognizing the novelty and importance of this work. Reviewers stated that "The authors present an intriguing idea and provide comprehensive measurement and calculations...", "offering a new perspective and approach", "the manuscript is interesting and timely", "the proposed technique... is potentially useful for nanoscale ferroelectric research", and "of interest to a large audience". We also appreciate the valuable and insightful comments and suggestions from reviewers, which further improve the quality of this work. In the revised version of the manuscript, we addressed each and every critical remark, as shown below.

Response to Reviewer #1:

In the study the authors demonstrate the nano-optical visualization of moiré domains in a twisted-WSe₂ bilayer. By placing a layer of graphene on top of the twisted-WSe₂ bilayer (TWB), they are able to observe distinct ferroelectricity in two different TWB domains using near-field optical microscopy. This is possible because the carrier densities, which determine the plasmonic responses, of graphene above the two distinct TWB domains differ due to the opposite electric polarizations in these ferroelectric domains. The authors present an intriguing idea and provide comprehensive measurements and calculations to support their conclusions, offering a new perspective and approach for studying nanoscale ferroelectricity in similar systems. I think the manuscript is interesting and timely. Here are a few concerns that I believe should be addressed by the authors prior to the publication of this manuscript.

We thank the reviewer for recognizing the novelty and interest of our work. We also appreciate the reviewer's comments and suggestions. We addressed each critical remark in the revised version of the manuscript.

1) This ferroelectric can be detected and imaged with other scanning probe microscopy such as piezoelectric force microscopy (PFM). What is the advantage of the method proposed in this work when compared with PFM? Introducing a graphene layer and back-gate to the bilayer and then image the sample with s-SNOM indeed can help obtain the local carrier density of different TWB domains, but in the meantime may also change the stacking order or twist angle of these domains (induced by strain effect when heating or annealing the sample) and also introduce other disturbing factors such as residues between different layers. The novelty of the current study should be stated more clearly.

We thank the reviewer for motivating us to elaborate on the advantages of the method proposed in this work. We totally agree with the reviewer that a graphene layer may change the stacking order or twist angle or introduce other changes/artifacts. We believe that the data we reported in the MS are not significantly influenced by these extrinsic factors. At the same time, our data show that integrating graphene with ferroelectric materials provides a wealth of information and an array of new opportunities. In the revised manuscript, we addressed the reviewer's suggestion by comparing s-SNOM with other methods and by stating more clearly the novelty of our study.

In the revised main text (page 3), we compared s-SNOM with other methods to illustrate the novelty of our work.

Piezoresponse force microscopy (PFM) and KPFM can directly measure the domain structures of ferroelectric via electromechanical surface deformation and electrostatic force, respectively. Whereas the ferroelectric domains can be clearly visualized by PFM (Ref.: Nature Nanotechnology 15, 580-584 (2020)), PFM results directly reflect the piezoelectric response, making it challenging to quantitatively evaluate ferroelectric properties. Usually, KPFM can

provide a quantitative characterization of a ferroelectric by measuring the work function. But KPFM has difficulty in quantitatively characterizing ferroelectric devices including multiple materials because disentangling the electrostatic force from individual layers is a formidable task.

In the revised main text (pages 12 and 13), we further highlight the novelty of the current study.

*Integrating the graphene layer with a ferroelectric allows us to explore how the ferroelectric tunes the properties of the proximal materials, which is crucial for the application of ferroelectric materials. We note that this proximal material is not limited to graphene. Recent advances in the nano-spectroscopy/imaging of excitonic effects in vdW materials set the stage for the analysis of the impact of ferroelectricity on the excitonic resonance energy and exciton diffusion, and for photovoltaic applications. Compared to PFM and KPFM, which are surface-sensitive methods, s-SNOM is a complementary method with the capability for tomographic imaging (Ref.: ACS Nano **8**, 6911-6921 (2014)). Therefore, s-SNOM enables one to probe hidden interfaces, for example, in devices with top electrodes. In the immediate future, our demonstrated method can be used to visualize the ferroelectric domain evolution under homogeneous electric fields in practical devices. This unique capability is essential for understanding the fundamental polarization dynamics.*

2) The measured carrier density (or Fermi level) of graphene is a key parameter for drawing conclusions in this paper. Therefore, the accuracy of deriving this number is of utmost importance. In the s-SNOM imaging, the domain wall of the twisted WSe₂ can reflect the tip-launched plasmon, and the domain sizes are comparable to the wavelength of graphene plasmons. Consequently, the experimentally obtained scattering amplitude may arise from the overall contribution of locally excited and reflected plasmons, similar to those in a graphene plasmonic cavity. This scenario could potentially interfere with the modulating of the doping level of graphene. How did the authors address or eliminate this possibility?

We thank the reviewer for bringing up the potential effect of plasmon polariton reflections on the extraction of carrier density. We totally agree with the reviewer that the domain wall of the twisted WSe₂ might reflect the tip-launched plasmon polaritons. If this reflection occurs, the measured scattering amplitude would arise from both locally excited and reflected plasmon polaritons. Fortunately, the reflected plasmon polaritons can only emerge when the carrier density is high enough for the plasmon polariton momentum to overlap with the momentum provided by tip or domain boundaries. Figure. R1a shows the plasmon polariton dispersion as a function of the carrier density. From Fig. R1a, we can see that the plasmon polariton momentum can only match the tip momentum until the doping reaches the regimes denoted by the purple shadow in the bottom right corner (the white dashed line denotes the center of the tip momentum). Furthermore, in the low doping regime, the plasmon polariton mode intensity, represented by $Im(r_p)$, is very weak. So for low-doped graphene, the near-field amplitude is dominated by the local excitation; if there is any launched and reflected plasmon, it cannot have a noticeable effect on the near-field amplitude. In summary, the reflected plasmon will not contribute to the measured scattering amplitude unless the doping is high.

As illustrated in Fig. R1b, the carrier density originating from a ferroelectric is extracted using the scattering amplitude data around the charge neutrality point (middle white region), while the propagating plasmon polaritons emerge at the regimes marked by the shadowed blue regions. It is

noteworthy that the horizontal shift between the two curves in Fig. R1b decreases at high doping ($V_g > 50$ V). We conjecture that this reduced shift is caused by the reflected plasmon polaritons, as pointed out by the reviewer. In the experiment, we also pay careful attention to make sure that we extract the carrier density from the near-field amplitude that was obtained at the regime without potentially reflected plasmon polaritons. Therefore, we believe we have minimized the graphene plasmonic effect in our data and analysis.

In the revised version, we further clarify the role of the plasmon reflection and explain how we address this potential issue, on page 5 of the main text and Supplementary Note 1 (page 3).

Fig. R1| **Plasmon polariton evolution as a function of doping.** **a**, The plasmon dispersion of Device B. The dispersion is visualized using a false-color map of the imaginary part of the reflection coefficient r_p (for the case of p-polarization, polarized along the tip). The white dashed line is an estimate of the momentum at which the coupling between the tip and plasmon polaritons is strongest. **b**, Representative results for the voltage dependence of the scattering amplitude, $s_4(V_g)$, probed within AB and BA domains.

3) The authors should state more clearly on how to obtain the carrier density in graphene at different gate voltage.

We thank the reviewer for his/her critical suggestion. We respectfully took the reviewer's suggestion by stating more clearly how to obtain the carrier density in graphene at different gate voltages in the Supplementary Materials, Note 1 (pages 2-3).

Here we summarize the key procedures for obtaining the carrier density. To obtain the carrier density, we need to know the charge density point of each domain and the geometric capacitance of the device.

1. To get the charge density point (CNP):

We recorded the near-field amplitude, s_4 , on each type of domain when the backgate voltage, V_g , was swept. Namely, we obtained s_4 versus V_g on two domains. The CNPs of the two types of domains correspond to the peak positions of s_4 in the s_4 versus V_g curves. We denote the CNPs of the AB and BA domains as V_{g-AB} and V_{g-BA} , respectively.

2. To calculate the carrier density of each domain:

a) The carrier density of the AB domain at backgate voltage V_g is $n_{AB} = \frac{C}{e}(V_g - V_{g-AB})$, where C is the geometric capacitance;

b) The carrier density of the BA domain at backgate voltage V_g is $n_{BA} = \frac{C}{e}(V_g - V_{g-BA})$;

c) The carrier density difference between the AB and BA domains is $\Delta n = \frac{C}{e}(V_{g-AB} - V_{g-BA})$. This carrier density difference originates from ferroelectricity.

4) In line 204, they wrote, 'but the ferroelectric polarization tuned Fermi energy makes these processes behave differently on two adjacent domains,' without specifying the exact different behavior exhibited by the two adjacent ferroelectric domains. Additionally, they cite Ref 41, which reported the visualization of defects on boron nitride, to support this statement. Please check if this is an appropriate citation.

We appreciate the reviewer for pointing out the possible confusion in our description of the responses of the two adjacent domains. The scenario we invoked here pertains to the possibility of charge transfer between graphene and WSe₂ (or h-BN). To eliminate the confusion, we have revised the manuscript. The revised sentences are shown below.

In our device, graphene is encapsulated by WSe₂ and h-BN. Defect states in WSe₂ and h-BN could result in charge transfer between WSe₂ (or h-BN) and graphene, a process governed by the Fermi energy⁴³. (Defects in h-BN and WSe₂ can be confirmed by optical measurements in Supplementary Note 10 and Supplementary Figure 17). Therefore, the ferroelectrically tuned Fermi energy of graphene may give rise to unequal charge transfer in the two neighboring domains. As a result, charge transfer can enhance the graphene doping beyond the ferroelectric potential (see Supplementary Note 5.5).

Regarding Ref. 41 (43 in the revised manuscript), this work demonstrated that the h-BN defects could be charged or discharged by tuning the graphene Fermi energy via tip gating. We used this citation to demonstrate that the charge transfer process depends on the Fermi energy of graphene. Thus, this is an appropriate citation. In addition, in the revised text, the citation was put at a more appropriate location.

5) In line 190, Page 7, they wrote "This is about four times higher than the theoretical prediction based on a linear screening model with ...". Is that a typing mistake? I could not find the exact information about this quantity relation, the authors are suggested to explain the statement more specifically.

We thank the reviewer for carefully reading the manuscript and pointing out a typo and potential confusion. In the revised version, the typo is corrected. The corrected sentence is "This value is about four times higher than the theoretically predicted 16 meV, based on a linear screening model with a ferroelectric potential of 56 mV (Supplementary Note 5)."

In addition, we respectfully took the reviewer's suggestion by more explicitly explaining the comparison between the measured doping and the theoretical prediction, on page 8 of the revised main text. All the revised sentences are shown below.

Around the CNP, the two scattering amplitude profiles acquired on the AB and BA domains, show a shift of $\Delta V_g = 12$ V (Fig. 2b). Based on the geometric capacitance of the device, we could translate the voltage into the charge density difference between adjacent domains, which is

$7.5 \times 10^{11} \text{ cm}^{-2}$. With the moiré-averaged Fermi energy at charge neutrality, this doping density shifts the Fermi energy (Fig.1b), measured with respect to the Dirac point in the two ferroelectric domains, by $\pm 71 \text{ meV}$. This value is about four times higher than the theoretically predicted 16 meV , based on a linear screening model with a ferroelectric potential of 56 mV (Supplementary Note 5).

6) In equation (1) the authors neglect the first term on the right-hand side, without explaining the reason.

We thank the reviewer for this suggestion. We respectfully took the reviewer's suggestion by elaborating on the photocurrent mechanism, particularly the first term of equation (1), on page 8 of the revised main text. The key points are summarized here.

By invoking the photo-thermoelectric effect and the Shockley-Ramo theorem, the collected photocurrent is expressed as:

$$I_{pc} = \int_{\Omega} \sigma S \nabla \delta T \cdot \nabla \psi d^2 \mathbf{r} = \int_{\partial \Omega} \sigma S \delta T \nabla \psi \cdot d \hat{\mathbf{n}} - \int_{\Omega} \sigma \delta T \nabla \psi \cdot \nabla S d^2 \mathbf{r} \quad (1)$$

where σ is the DC conductivity, S is the Seebeck coefficient, δT is the increased electron temperature, and ψ is an auxiliary field obtained by assigning potentials of 1 and 0 to contacts. The first term on the right-hand side of equation (1) denotes the photocurrent formed at the contact edges. To form a nonzero photocurrent at the edges, the edge regions must have nonzero conductivity, nonzero Seebeck coefficient, and an increased electron temperature. However, when the measured moiré region is far away from the contacts, farther than the cooling length, the tip-enhanced local field cannot heat up the electron temperature near the contacts; i.e., $\delta T=0$. Therefore, the photocurrent formed at the edge is zero, and we can safely ignore this term.

Response to Reviewer #2

In this paper, the authors studied ferroelectricity in minimally twisted WSe2 by using near-field infrared nano-imaging and nano-photocurrent measurements. They claimed that the plasmonic response in a graphene layer adjacent to the moire WeS2 bilayers could be used to unveil the local ferroelectric polarization. The proposed technique is able to resolve ferroelectric domains in nanoscale, which should be potentially useful for nanocale ferroelectricity research for optoelectronic devices. The paper is well-written and could be considered publication in Nat. Commun. as long as following issues are considered.

We thank the reviewer for acknowledging the importance of this work, recommending publication, and providing his/her valuable comments and suggestions.

1. First, the near-field amplitude instead of plasmon wavelength is used to determine the carrier density of the graphene in this work. However, the near-field amplitude is affected by lots of facts, such as the tip, laser intensity and so on. Could the authors comments on further how to guarantee the quantized accuracy?

We thank the reviewer for their question on the accuracy of the measured carrier density. We totally agree with the reviewer that the near-field amplitude is affected by many factors, such as tip geometry and laser intensity. The presence of all of these experimental parameters would make it challenging to extract the accurate carrier density from the near-field amplitude. On the other

hand, plasmon wavelength is another parameter that is frequently used to extract the carrier density. However, in moiré samples, it is difficult to extract the plasmon polariton wavelength accurately. This is because plasmon polaritons form complex patterns due to reflection by the domain boundaries, unlike the clear periodic patterns that emerge when imaging polaritons near a homogeneous sample edge.

In order to extract the carrier density and guarantee its accuracy, neither the plasmon fringe period nor the absolute scattering amplitude was used. Instead, we used the peak positions of the near-field amplitude for carrier density analysis. We found that the near-field amplitude peaks at the charge neutrality point (CNP) of graphene. Therefore, we can precisely read out the CNP positions of each domain when tuning the backgate. From the difference in CNP between domains with opposite polarization, we can get their doping contrast from ferroelectricity. In addition, with the knowledge of each domain's CNP and the devices' capacitor geometry, we can get the carrier density of each domain at various backgate voltages.

We respectfully took the reviewer's suggestion by elaborating on how to extract the accurate density in Supplementary Materials, Note 1 (pages 2-3).

2. In Fig.3, some data comes from Device B instead of Device A, what's the difference in physical properties between Device A and B? Besides, it is a bit confused to compare the result comes from different samples.

We thank the reviewer for their comment. We used data for Device B to show the photocurrent evolution. Devices A and B have identical architecture, and the two devices display consistent properties (see, for example, Fig. 2 for Device B and Fig. S14 for Device A). For Device A, we measured a series of photocurrent/nano-IR images, with a backgate voltage step of 5 V. We collected a detailed series of images for this device, yet our backgate dependence data are only fragmentary. We acquired this additional information for Device B. The data we show for Device B are in the form of a line scan, with a fine backgate step size of 0.8 V. These latter measurements provided systematic information on the backgate dependence. In order to show more comprehensive photocurrent evolution data to readers, we chose to show data on Device B in panel e.

To address the reviewer's concern and eliminate the potential for confusion, in the caption of Fig. 3 we carefully explained why panel e shows data on Device B.

3. The infrared laser with a frequency of 880 cm^{-1} is used in this work, is there any special reason for the choice?

We thank the reviewer for bringing up the excitation frequency. In the main text, all the data were acquired using frequencies around 880 cm^{-1} . However, there is no particular reason for this choice. In principle, any lasers with frequencies that can excite the plasmon polaritons of graphene, can be used in our work. We see from the plasmon dispersion that we can choose laser frequencies from THz to middle infrared with frequencies around 2000 cm^{-1} .

In order to demonstrate that other excitation frequencies can be used to probe the ferroelectricity, we also performed measurements using various frequencies, such as 861 cm^{-1} , 888 cm^{-1} , and 920 cm^{-1} , as shown in Fig. R2. We can see that the observed ferroelectric features are very similar for the various excitation frequencies. The slight differences between them originate from the energy

dependence of the plasmonic response. Incidentally, the wavelength dependence further confirms that the contrast between domains originates from the plasmonic response.

In the revised version, we respectfully took the reviewer's suggestion by adding more discussion of the excitation frequency to Supplementary Note 8, titled "Photon energy dependence of the ferroelectric modulated plasmonic response."

Fig. R2| **Moiré ferroelectric probed through plasmonic response using lasers with various wavelengths.** **a,b,c,** The near-field scattering amplitude as a function of gate voltage $V_g - V_{\text{CNP}}$, measured along a line trace that crosses several domains (marked with the white line in **d**). **d,** Image of the near-field scattering amplitude s_4 at excitation energy $\omega = 888 \text{ cm}^{-1}$ and $V_g - V_{\text{CNP}} = 125 \text{ V}$. **e,** The evolution of the plasmonic response as a function of global backgate voltage probed at various photon energies. The arrows in panels **a**, **b**, and **c** indicate where the line profiles are taken. As indicated by the arrow, the plasmonic resonance shifts to higher carrier density with increasing photon energy.

4. In Fig.1e, an obvious different variation trend exists for the case of 90V compared with that of 10V and 150V, what's the possible reason for the difference?

We thank the reviewer for carefully reading the manuscript and pointing out the different spatial variation trends at various backgate voltages. These different trends are rooted in the non-monotonic evolution of near-field scattering amplitude as a function of carrier density, as shown in Fig. R3b. At $V_g - V_{\text{CNP}} = 90 \text{ V}$, the two neighboring domains have different carrier densities, but the near-field scattering amplitudes are almost the same. Therefore, from the linecut of the near-field scattering amplitude, we can see that the period doubles at $V_g - V_{\text{CNP}} = 90 \text{ V}$. This

period-doubling feature can be clearly identified by images in Fig. R3a and the corresponding linecut in Fig. R3c.

In the revised main text (page 9), we explain these different variation trends at different backgate voltages.

Fig. R3| The evolution of ferroelectrically tuned plasmon response as a function of spatial position and gating voltage. a, Nano-infrared images acquired for different gate voltages with excitation energy $\omega=888\text{ cm}^{-1}$. **b,** Voltage-dependence of the scattering amplitude, $s_4(V_g)$, probed within AB and BA domains. These two traces were acquired at positions marked by “x” in the top panel of **a**. **c,** Line profile of the scattering amplitude, s_4 , for selected backgate voltages measured along the line indicated in Panel **a**. Green dashed lines denote the positions of the domain walls.

5. Due to the complex doping mechanism for the graphene, a quantitative measurement of the ferroelectricity of the twisted WSe2 may be needed, which can verify the correctness of the explanation for experimental results.

We thank the reviewer for their valuable suggestion. As the reviewer stated, the doping mechanism for graphene is complex. In order to verify the correctness of our conclusions, we studied multiple devices, and consistent results were obtained. In addition, the ferroelectrically induced carrier density in graphene was demonstrated by putting graphene on an oxide ferroelectric material (Ref.: *Nature Communications* **6**, 6136 (2015)). Recent electric transport measurements also indicated that graphene is doped by adjacent two-dimensional ferroelectrics (Ref.: *Science* **372**, 1458-1462 (2021); *Nature nanotechnology*, **17**, 367-371 (2022)). Moreover, nano-IR is a reliable approach to measuring the ferroelectrically induced carrier density (Ref.: *Nano letters* **15**, 4859-4864 (2015)).

PFM and KPFM are widely used to characterize ferroelectric materials. For the purpose of a quantitative characterization of ferroelectricity, we performed the measurements using both PFM and KPFM. The ferroelectrics can be clearly visualized by PFM, as shown in Figs. S1,2. However, PFM results directly reflect the piezoelectric effect, making it challenging to quantitatively evaluate ferroelectric properties, especially in our devices that feature integration with back-gated graphene. Usually, KPFM can provide a quantitative characterization of a ferroelectric by measuring the work function. However, in our devices, underneath the WSe2 ferroelectric domains, there is graphene, with a different work function (or Fermi energy). The graphene is also involved in the KPFM response. Consequently, with the current complex device structure, it is challenging

to read out the pure ferroelectric potential of WSe₂. Quantitatively characterizing the ferroelectric is an essential research topic and can be explored in future studies.

In the revised manuscript, we addressed the reviewer's suggestion by further elaborating on the ferroelectrically induced doping (page 5) and by adding a discussion of the quantitative measurements of ferroelectricity (page 3).

Response to Reviewer #3

The authors reported the observation of ferroelectric domains in twisted WSe₂ using near-field infrared nano-imaging and nano-photocurrent techniques. This is achieved because of the fact that the plasmonic response of monolayer graphene depends on its carrier density, which is turned by the polarization of the twisted WSe₂. The work is of interest to a large audience, and I recommend publication of the manuscript after the following minor concerns are clarified.

We thank the reviewer for his/her positive remarks and for recommending the publication of our work. The reviewer's comments and suggestions are appreciated.

1, Regarding Fig. 2b, a qualitative description in the main text on why would the near-field amplitude first decreases and then increases would be helpful for the general audience.

We thank the reviewer's excellent suggestion of qualitatively describing the near-field amplitude evolution in the main text. We respectfully took the reviewer's suggestion by adding a description of why the near-field amplitude first decreases and then increases. Below, we briefly summarize the mechanism of near-field amplitude evolution.

The near-field amplitude is governed by the Fresnel coefficient of p-polarized light, $r_p(\omega, q)$, which is a function of the plasmon polariton energy, ω , and momentum, q (Refs.: *Physical Review B* **85**, 075419 (2012); *Physical Review B* **90**, 085136 (2014)). At low carrier density, $Amp(r_p(\omega, q))$ decreases with doping, and thus the near-field amplitude decreases. (More details on why $Amp(r_p(\omega, q))$ decreases with doping, are provided in Supplementary Materials, Note 3.2.) When the carrier density, n , is further increased, the plasmon polariton momentum gradually decreases, obeying $\sqrt{q} \propto \frac{\omega}{\sqrt{n}}$, and finally matches the tip momentum. Consequently, $Amp(r_p(\omega, q))$ tremendously surges, resulting in an upturn of the near-field amplitude. The momentum overlaps between the plasmon polariton and the tip, result in a better coupling between them, and thus the tip can efficiently launch propagating plasmon polaritons. Therefore, the increase in near-field amplitude is followed by the emergence of plasmon polariton interference fringes.

In the revised main text (page 7), we added a qualitative description of near-field amplitude evolution.

2, Regarding the contrast between neighboring domains, the authors invoke interfacial charge trapping and defect states in WSe₂ or h-BN to explain the larger-than-expected contrast. This is a bit confusing. If one assumes these trappings or defects distribute uniformly across the samples, shouldn't they contribute similarly to the backgate, as in shifting the CNP of both domains in the

same direction? Can the authors elaborate on how would they increase the contrast between domains?

We thank the reviewer for pointing out the confusion. The charge transfer between defect states and graphene, depends on their energy band alignment. The ferroelectric potential results in a relative Fermi energy shift between two neighboring graphene domains. Therefore, even though the defects distribute uniformly across the samples, the charge transfers in domains with different graphene energy bands cannot be the same (more discussion of charge transfer can be found in Supplementary Note 5.5). To eliminate the confusion, we have revised the manuscript (page 8). The revised sentences are shown below.

In our device, graphene is encapsulated by WSe₂ and h-BN. Defect states in WSe₂ and h-BN could result in charge transfer between WSe₂ (or h-BN) and graphene, a process governed by the Fermi energy⁴³. (Defects in h-BN and WSe₂ can be confirmed by optical measurements in Supplementary Note 10 and Supplementary Figure 17). Therefore, the ferroelectrically tuned Fermi energy of graphene may give rise to unequal charge transfer in the two neighboring domains. As a result, charge transfer can enhance the graphene doping beyond the ferroelectric potential (see Supplementary Note 5.5).

3, Regarding the photocurrent, due to the ferroelectric polarization, the Fermi levels of the graphene beneath neighboring domains are different and a potential difference naturally develops across the domain wall, will this lateral potential variation produce photocurrent? Is it absolutely necessary to introduce Seebeck effect?

We thank the reviewer for his/her remark on the photocurrent mechanism. We absolutely agree with the reviewer's statement that a potential difference naturally develops across the domain wall. However, the photocurrent originating from this potential can be ignored, for the following reasons. First, in graphene with a potential junction, the photocurrent from the photovoltaic effect is much weaker than that from the Seebeck effect, which has been well investigated (Refs.: *Nano Lett.* **10**, 562-566 (2010), *Science* **334**, 648 (2011)). Second, the backgate dependence of photocurrent in our experiment shows that the photocurrent reaches a maximum near the CNP, where the Seebeck coefficient contrast between two neighboring domains is maximized. This dependence indicates that the photocurrent arises from the Seebeck effect. Third, in our nano-photocurrent experiment, the decay length of photocurrent at the domain wall is hundreds of nanometers, which is much larger than the potential junction width of ~10 nm. This extraordinary observed decay length is caused by the large cooling length of the hot electrons in graphene.

To summarize, the photocurrent originating from the potential at the domain wall is expected to show a fast spatial decay. Conversely, a much slower decay was observed in our nanometer-resolved photocurrent mapping. Our nano-photocurrent results, in concert with previous far-field measurements, corroborate that the photocurrent in graphene is dominated by the photothermal effect. Therefore, it is absolutely necessary to introduce the Seebeck effect.

In the revised version, in order to address the effect of this voltage potential on the photocurrent, we added a new subsection titled "Photovoltaic effect in graphene photocurrent" to the Supplementary Materials (Note 11.2, page 29).

4, How large is the laser spot as compared to the moire pattern in the nano-photocurrent measurements?

We thank the reviewer for his/her important question about spatial scale in the nano-photocurrent measurement. This question motivates us to revise the manuscript, in order to explain the nano-photocurrent formation more clearly. Below, we describe all the spatial scales relevant to our nano-photocurrent measurements.

The laser with a wavelength of ~ 11 μm is focused on the sample and tip using a parabolic mirror; the laser spot diameter on the sample is ~ 30 μm . It should be noted that this incident light is locally enhanced by the sharp metallized tip. The nano-photocurrent, acquired by demodulation at the tip-tapping frequency, is induced by the locally enhanced field at the apex of the tip. Therefore, to analyze the nano-photocurrent, the more relevant spatial scale is the size of the locally enhanced field, rather than the laser spot size. This local field is confined to \sim tens of nanometers underneath the tip and is much smaller than the moiré period, which is around hundreds of nanometers.

In the revised version, we added a new subsection titled “Spatial scales in graphene photocurrent” to the Supplementary Materials (Note 11.3, page 29), in order to describe all the relevant spatial scales in the photocurrent measurements.

REVIEWERS' COMMENTS

Reviewer #1 (Remarks to the Author):

The authors addressed the comments raised by reviewers well. The manuscript can be accepted as is.

Reviewer #2 (Remarks to the Author):

In the revised paper, my comments are fully considered by the authors and the quality of the paper is improved. As a result, I recommend its publication in Nat. Commun.

Reviewer #3 (Remarks to the Author):

The authors have addressed the issues raised in my previous review, and I have no further comments.